# Construction of a Full-Length Infectious Clone Derived from Type O Foot-and-Mouth Disease Virus Isolated in South Korea for Vaccine Development with High Antigen Productivity

**DOI:** 10.3390/vaccines13121195

**Published:** 2025-11-26

**Authors:** Jae Young Kim, Sun Young Park, Gyeongmin Lee, Sang Hyun Park, Jong Sook Jin, Jong-Hyeon Park, Young-Joon Ko

**Affiliations:** Center for Foot-and-Mouth Disease Vaccine Research, Animal and Plant Quarantine Agency, 177 Hyeoksin 8-ro, Gimcheon-si 39660, Gyeongsangbuk-do, Republic of Korea; ivorikim@korea.kr (J.Y.K.); sun3730@korea.kr (S.Y.P.); lgm6004@korea.kr (G.L.); shpark0205@korea.kr (S.H.P.); in75724@korea.kr (J.S.J.); parkjhvet@korea.kr (J.-H.P.)

**Keywords:** foot-and-mouth disease, type O, infectious clone, South Korea, vaccine

## Abstract

Background: Foot-and-mouth disease (FMD) is a highly contagious viral disease of cloven-hoofed animals such as cattle and pigs, characterized by vesicular lesions in the mouth, nose, teats, and feet. Globally, the most commonly used FMD vaccines are inactivated vaccines produced by chemical inactivation of the infectious FMD virus (FMDV). This study aimed to establish an infectious clone of the O/Boeun/SKR/2017 virus that has demonstrated the highest antigen productivity among the various type O vaccine strains developed in South Korea to date. Methods: An infectious clone was generated from a type O virus isolated during the 2017 FMD outbreak in South Korea. The viral genome was divided into two fragments, each amplified separately, and subsequently ligated to produce a full-length infectious clone. Results: Rescue of infectious FMDV was confirmed using a commercial antigen detection kit and electron microscopy. Under optimized culture conditions, the rescued virus titer reached 2 × 10^7^ TCID_50_/mL, and the antigen yield was 6.4 µg/mL. Following inactivation, the antigen was formulated into a vaccine and administered to pigs. Four weeks post-vaccination, challenge with the live virus resulted in no clinical symptoms, demonstrating complete protective efficacy. Conclusions: To the best of our knowledge, this is the first report describing the construction of an infectious clone derived from a field FMDV isolate in South Korea and its application in vaccine development. The O/Boeun/SKR/2017 infectious clone may serve as a genetic backbone for the rapid generation of new FMD vaccine candidates with high antigen productivity by substituting epitopes from other FMDV.

## 1. Introduction

Foot-and-mouth disease (FMD) is a viral disease that affects cloven-hoofed animals, such as cattle, pigs, and goats, causing vesicular lesions in the mouth, nose, teats, and feet [1]. As FMD can also be transmitted through aerosols, rapid culling or vaccination is essential to prevent the further spread of the disease [2]. The causative agent, the foot-and-mouth disease virus (FMDV), is a non-enveloped, spherical virus containing a single-stranded, positive-sense RNA genome enclosed within a protein capsid. The structural proteins VP1–VP4 assemble into protomers, and the five protomers combine to form a pentamer. Twelve pentamers were assembled into viral precursor particles. During genome encapsidation, VP0 is cleaved into VP2 and VP4, resulting in the formation of the mature virion, known as the 146S particle [3]. The non-structural proteins responsible for viral replication and processing include L, 2A, 2B, 2C, 3A, 3B, 3C, and 3D [4]. FMDV is classified into seven serotypes: O, A, C, Asia1, SAT1, SAT2, and SAT3. There are multiple topotypic variants within each serotype. Serotype O is the most prevalent worldwide, whereas serotype A exhibits the greatest antigenic diversity with numerous subtypes. The Asia1 serotype has mainly been reported in Asian countries. The C serotype has not been reported worldwide since 2004. SAT serotypes previously confined to Africa have recently spread to the Middle East [5,6,7,8,9,10].

In Korea, 13 FMD outbreaks were reported between 2000 and 2025. Following a large-scale outbreak in 2010–2011, which caused severe economic losses, nationwide vaccination of all cloven-hoofed animals was implemented [11]. However, all FMD vaccines currently used in Korea are imported, posing challenges to a stable supply. Therefore, efforts are underway to establish domestic FMD vaccine production facilities and develop indigenous vaccine strains and manufacturing technologies.

The most widely used FMD vaccine worldwide is the inactivated vaccine produced by chemically inactivating infectious viruses [12]. Field-isolated viruses frequently demonstrate low antigen productivity in cell culture, which can render them unsuitable as vaccine seed strains. To address these challenges, the construction of infectious cDNA clones from field isolates offers an opportunity for targeted genetic modifications, thereby enhancing viral characteristics such as antigen yield and stability. Consequently, the development of inactivated vaccines based on these infectious clones presents a significant advantage over traditional vaccines derived directly from field strains [13].

In this study, an infectious clone was constructed using the O/Boeun/SKR/2017 virus, which showed the highest antigen productivity among several candidate in-house vaccine strains. Antigen production and protective efficacy were evaluated in pigs to determine the suitability of the virus as a vaccine. If this infectious clone is effective, it could serve as a useful platform for developing FMD vaccines with high antigen productivity.

## 2. Materials and Methods

### 2.1. Cells and Viruses

ZZ-R fetal goat tongue cells (friedrich-Loeffler-Institute, Riems, Germany) were routinely maintained in Dulbecco’s modified Eagle medium/Ham’s F12 medium (Corning, Union City, NJ, USA). Baby hamster kidney (BHK-21) adherent cells (C-13, ATCC CCL-10, Manassas, VA, USA) were maintained in DMEM (Corning) containing 10% fetal bovine serum (FBS; Gibco, Paisley, UK) and 1% antibiotic–antimycotic solution (Gibco). The cultures were incubated at 37 °C in a humidified atmosphere with 5% CO_2_. Suspension-adapted BHK-21 cells, which were co-developed by the Animal and Plant Quarantine Agency (APQA) and the Korea Research Institute of Bioscience and Biotechnology (KRIBB), were grown in serum-free Cellvento™ BHK-200 medium (Merck, Darmstadt, Germany) at 37 °C and 110 rpm in an orbital shaker. Cell density and viability were determined by trypan blue exclusion assay using a Vi-Cell XR automated analyzer (Beckman Coulter, Brea, CA, USA).

For viral infection, approximately 70% of the total culture volume (3 × 10^5^ suspension-adapted BHK-21 cells/mL) was grown for 3.5 d to reach about 3 × 10^6^ cells/mL, after which 30% fresh Cellvento medium was added without removing the used medium. The O/Boeun/SKR/2017 virus (GenBank accession no. MG983730.1), isolated during outbreaks in South Korea, was used to inoculate BHK-21 suspension cultures at a multiplicity of infection (MOI) of 0.001. The virus-infected cultures were harvested 16 h post-infection (hpi), and the supernatants were collected by centrifugation (4000× *g*, 20 min, 4 °C) for subsequent viral titration and quantification of 146 S particles. The viruses were treated with 3 mM binary ethylenimine (Sigma-Aldrich, St. Louis, MO, USA) at 26 °C for 28 h. The reaction was subsequently quenched by adding 1 M sodium thiosulfate (Daejung Chemicals & Metals, Siheung, Republic of Korea) to achieve a final concentration of 2% (*v*/*v*).

### 2.2. Cloning of Full-Length cDNA

This procedure was conducted by replacing the full-length O1 Manisa genome in the pre-constructed pBluescript II SK (+) plasmid vector with the full-length O/Boeun/SKR/2017 genome. The viral genome was divided into two fragments (small and large), and each fragment was amplified using PCR. Viral RNA of the O/Boeun/SKR/2017 virus was extracted from the culture supernatant of virus-infected BHK-21 cells using a QIAamp Viral RNA Mini Kit (Qiagen, Hilden, Germany), and the final elution was performed with RNase-free water. Reverse transcription was performed using SuperScript IV Reverse Transcriptase (Invitrogen, Carlsbad, CA, USA) with an oligo(dT) primer annealed to the 3′ terminal poly(A) region of the viral genome. The reaction was performed at 42 °C for 60 min, followed by enzyme inactivation at 70 °C for 5 min. The synthesized cDNA was used as a template for PCR amplification. First, for the small fragment, PCR was performed using a primer set designed for the O/Boeun/SKR/2017 cDNA, in which an NdeI restriction site was introduced at the 5′ end, and an AvrII restriction site was introduced at the 3′ end for substitution of guanine for adenine at 375th of the template. Primers were designed to amplify the viral genome fragment from the NdeI site located upstream of the T7 promoter in the pBluescript II SK(+) vector, to the AvrII site downstream of the poly(C) track. PCR amplification was performed using Phusion High-Fidelity DNA Polymerase (Thermo Fisher Scientific, MA, USA) in a 50 µL reaction volume under the following conditions: an initial denaturation at 98 °C for 30 s; 30 cycles of 98 °C for 10 s, 58 °C for 30 s, and 72 °C for 30 s per kb; followed by a final extension at 72 °C for 10 min. The PCR products were purified using a PCR Clean-Up Kit (Macherey-Nagel, Düren, Germany), quantified, and verified by 1% agarose gel electrophoresis. The resulting PCR product was inserted into the pBluescript II SK (+) plasmid vector containing the O1 Manisa genome and digested with the same restriction enzymes (NdeI and AvrII). The pBluescript II KS(+) vector containing the full-length O1 Manisa genome was digested with NdeI and AvrII to isolate the backbone fragment. The digested DNA was separated by agarose gel electrophoresis, and the backbone DNA was recovered by gel extraction. The purified backbone and the small fragment PCR product of the O/Boeun/SKR/2017 virus were mixed at a molar ratio of 1:5 and ligated using Ligation High ver.2 (Toyobo, Osaka, Japan) at 16 °C for 30 min. Next, for the large fragment, PCR was performed using the O/Boeun/SKR/2017 virus cDNA as a template, with primers introducing an AvrII site at the 5′ end and a NotI site at the 3′ end. To obtain the large fragment of the O/Boeun/SKR/2017 virus, primers were designed to include an AvrII site at the 5′ end and a NotI site at the 3′ end following the 3′ UTR. PCR amplification was performed with Phusion High-Fidelity DNA Polymerase in a 50 µL reaction volume under the following conditions: an initial denaturation at 98 °C for 30 s; 30 cycles of 98 °C for 10 s, 60 °C for 20 s, and 72 °C for 30 s per kb; followed by a final extension at 72 °C for 10 min. The amplified product was purified and verified by agarose gel electrophoresis, as described above. Finally, in the pBluescript II SK (+) plasmid vector containing the previously inserted O/Boeun/SKR/2017 viral small fragment, the O1 Manisa genome large fragment was replaced with a large fragment of the O/Boeun/SKR/2017 genome obtained by digestion with AvrII and NotI. A plasmid containing a small fragment of the O/Boeun/SKR/2017 genome substituted into the O1 Manisa backbone was amplified in *Escherichia coli* and extracted. The large fragment of the O1 Manisa genome was excised by double digestion with AvrII and NotI. The purified O/Boeun/SKR/2017 large fragment PCR product was then ligated into the backbone at a molar ratio of 1:5 using a T4 DNA Ligase Kit (New England Biolabs, Ipswich, MA, USA) at 16 °C for 30 min. Table 1 lists the primer sequences used for recombinant virus construction.

### 2.3. Rescue of FMDV Through Transfection in Serial Cell Lines

The plasmid was purified using the Nucleo Spin Plasmid Kit (MACHEREY-NAGEL, Düren, Germany). Following purification, plasmid concentration and purity were assessed using a DeNovix DS-11 Fx+ spectrophotometer (DeNovix, Inc., Wilmington, DE, USA). The results showed an A260/280 ratio of 1.85 and an A260/230 ratio of 2.18, confirming that the plasmid DNA was of high purity with no detectable protein or organic compound contamination. The purified plasmid was then transfected into BHK/T7-9 cells expressing T7 RNA polymerase using Lipofectamine 3000 (Invitrogen, Waltham, MA, USA), according to the manufacturer’s protocol. For transfection, 1 × 10^6^ BHK/T7-9 cells were seeded in a 6-well plate, and 2 µg of the O/Boeun/SKR/2017 expression vector was applied. The cells were incubated at 37 °C in 5% CO_2_ for three days, and the virus was harvested after freezing and thawing. The rescued viruses were subsequently inoculated into ZZ-R cells and passaged until cytopathic effects were observed. The virus recovered from these cultures was sequentially propagated in adherent BHK-21 cells and suspension BHK-21 cells. Rescued FMDV was confirmed using an antigen detection rapid test kit (VDRG FMDV 3Diff/Pan Ag Rapid kit, Median Diagnostics, Chuncheon, Kangwon-do, Republic of Korea), which is a lateral flow chromatographic immunoassay for the universal detection of all seven serotypes of FMDV antigens and simultaneous typing of O, A, and Asia1 in saliva and sera.

### 2.4. Transmission Electron Microscopy

Viral suspensions concentrated by polyethylene glycol (PEG) precipitation were layered onto 15–45% sucrose density gradient tubes and centrifuged at 100,000× *g* for 4 h. The interphase between the 30% and 35% sucrose layers was collected and subjected to a second ultracentrifugation step at the same speed and duration. The resulting pellet was resuspended and dialyzed against 50 mM Tris buffer containing 300 mM KCl (pH 7.6) at 4 °C to remove sucrose. A small droplet of the purified virus was placed on Formvar-coated copper grids, negatively stained with 1% uranyl acetate, and examined under a transmission electron microscope (H-7100FA; Hitachi, Tokyo, Japan).

### 2.5. Virus Titration

Infectious viral titers were measured in adherent BHK-21 monolayers using an endpoint dilution. The titers were expressed as the 50% tissue culture infectious dose (TCID_50_/mL), which was calculated per the Spearman–Kärber statistical method [14].

### 2.6. Quantification of FMDV Particles

The FMDV particle-quantification method is based on the results described in previous studies [15,16]. Viral culture supernatants were mixed with chloroform (1:1, *v*/*v*; Merck KGaA, Darmstadt, Germany) and vigorously inverted for 5 min. Following centrifugation (3000× *g*, 15 min, 4 °C), the aqueous layer was collected and re-extracted with chloroform once more. The samples were then centrifuged at 16,000× *g* for 10 min, and the resulting supernatants were treated with benzonase (Sigma-Aldrich) at 0.025 U/μL and incubated with agitation at 37 °C for 1 h. After a final centrifugation (16,000× *g*, 10 min, 4 °C), the clarified supernatant was filtered through 0.22 μm Millex-GV filters (Merck KGaA, Darmstadt, Germany). Quantification of intact 146S particles was performed using size-exclusion high-performance liquid chromatography (SE-HPLC) with a TSKgel G4000PWXL column (300 mm × 7.8 mm) coupled to a PWXL Guardcol column (Tosoh Bioscience, Tokyo, Japan) on an Agilent 1260 Infinity II system (Agilent Technologies, Santa Clara, CA, USA). The mobile phase consisted of 30 mM Tris-HCl and 400 mM NaCl (pH 8.0) at a flow rate of 0.5 mL/min. Chromatographic data were processed using the OpenLAB CDS ChemStation software (version 3.2.0.620), and 146 S particle concentrations (µg/mL) were calculated following the method described previously [17].

### 2.7. Animal Experiment

The monovalent vaccine was composed of the rescued FMDV (15 μg per dose), 1% saponin (Sigma-Aldrich), and 1% aluminum hydroxide gel (General Chemical, Mount Laurel, NJ, USA). The ISA 206 VG adjuvant (Seppic, Paris, France), pre-equilibrated to 30 °C, was mixed with the antigen preparation at a 1:1 ratio to generate an experimental vaccine with a final volume of 2 mL per dose. The formulations were placed in a water bath at 20 °C for 1 h without light exposure and were then stored at 4 °C until administration. Eight two-month-old pigs, confirmed to be seronegative for FMDV, were used in the animal experiment. Five animals were immunized intramuscularly with 2 mL of the test vaccine, whereas three pigs served as unvaccinated controls. A minimum of three animals is required in both the experimental and control groups to allow for basic statistical analysis; therefore, the negative control group was set at three animals. In vaccine studies, unexpected mortality or dropout may occur due to stress during vaccination or virus challenge, as well as individual variability. Because the loss of even a single animal in the experimental group can substantially affect the statistical interpretation, the vaccination group was composed of five animals to provide an adequate margin. Blood samples were collected on days 0, 7, 14, 21, and 28 postvaccination (dpv). At 28 dpv, all pigs were challenged with O/Boeun/SKR/2017 at 1 × 10^5^ TCID_50_ per 0.1 mL via heel bulb injection. Animals showing clinical signs of infection were promptly isolated to prevent secondary spread. Clinical scoring followed the criteria of Alves et al. [18]: appetite loss (1–2 points); lameness (1–2), coronary band pain (1–2), foot vesicles (up to 4 points); and oral lesions on the tongue, gums, lips, or snout (up to 3 points), with a maximum score of 13. Protection was defined as the absence of visible lesions anywhere except at the injection site for seven days after the challenge. Virus shedding in nasal secretions and sera was monitored daily for 1 week post-infection using real-time RT-PCR (AccuPower FMDV Real-Time RT-PCR Master Mix Kit; Bioneer, Daejeon, Republic of Korea). Animal experiments were approved by the Institutional Animal Care and Use Committee (IACUC) of the Animal and Plant Quarantine Agency (IACUC No. 2024-831) on 8 February 2024.

### 2.8. Virus Neutralization Test

Neutralizing antibody titers against FMDV were measured according to World Organization for Animal Health Terrestrial Manual guidelines [19]. Serum samples were heat-inactivated at 56 °C for 30 min before testing. Starting from a 1:8 dilution, sera were serially twofold-diluted in duplicate (50 µL per well) and mixed with 100 TCID_50_ of virus. After 1 h of incubation at 37 °C, porcine kidney (LFBK) cells (0.5 × 10^6^ cells/mL; Plum Island Animal Disease Center, Orient, NY, USA) were added. Plates were sealed and incubated for 2–3 d at 37 °C in 5% CO_2_. Virus-neutralizing (VN) titers were expressed as log_10_ reciprocals of the highest serum dilution that completely neutralized 100 TCID_50_ of virus, calculated using the Spearman–Kärber formula [14]. In general, a VN titer of 1/45 or more of the final serum dilution is regarded as positive. However, for certification of an individual animal for the purpose of international trade, a titer of 1/16 is considered to be positive [19].

### 2.9. Enzyme-Linked Immunosorbent Assay

Type O FMDV-specific antibodies were quantified by competition ELISA using the PrioCHECK FMDV Type O kit (Prionics, Lelystad, The Netherlands) according to the manufacturer’s instructions. Serum samples were incubated on antigen-coated plates for 1 h at 25 °C, washed, and then treated with horseradish peroxidase-conjugated monoclonal antibodies against FMDV structural proteins. The optical density was measured at 450 nm after substrate development. The ELISA results were interpreted based on percentage inhibition, with values ≥ 50% considered positive and those < 50% considered negative.

### 2.10. Statistical Analysis

All data were expressed as mean ± standard deviation (SD) from three independent experiments. Statistical differences were evaluated by one-way analysis of variance (ANOVA), followed by Tukey’s honest significant difference (HSD) post hoc test using GraphPad Prism 5 (GraphPad Software, San Diego, CA, USA). Groups with different letters were considered significantly different (*p* < 0.05).

## 3. Results

### 3.1. Construction of an Infectious Clone Derived from the Domestic FMDV

The entire viral genome was divided into two fragments, each amplified by PCR, and ligated together (Figure 1A). A small fragment of the O/Boeun/SKR/2017 genome was amplified by PCR, and the expected 415 bp band was confirmed by agarose gel electrophoresis (Figure 1B). The plasmid containing the full-length FMDV O1 Manisa genome was digested with AvrII, revealing a corresponding size of about 11 kb (Figure 1C). A large fragment of FMDV O/Boeun was amplified using PCR, yielding a band of approximately 7.8 kb (Figure 1D). This fragment was ligated into a pBluescript II SK(+) vector containing a small fragment of the O/Boeun/SKR/2017 genome. Double digestion of the resulting recombinant plasmid with AvrII and NotI produced a large fragment and the plasmid containing a small fragment (Figure 1E). Full-length sequencing analysis was performed, and sequence comparison of the infectious clone derived from the O/Boeun/SKR/2017 genome with the previous vaccine strain, O/SKR/Boeun/2017, revealed nearly identical amino acid sequences, except for a single variation from Serine to Glycine at the 193rd residue in the VP2 region.

### 3.2. Rescue of Infectious FMDV from the cDNA Clone

Following transfection of the plasmid into BHK/T7-9 cells, the collected supernatant was used to inoculate ZZ-R cells, resulting in the appearance of cytopathic effects. The recovered virus was subsequently propagated in both adherent and suspension BHK-21 cells, after which the culture supernatant containing viral particles was harvested. The viral titer of the harvested supernatant was determined by TCID_50_ assay, yielding a titer of 1.28 × 10^8^ TCID_50_/mL. The virus supernatant was tested using a commercial antigen detection kit, resulting in a positive band corresponding to the type O FMDV (Figure 2A). Purified viral particles were subsequently visualized under a transmission electron microscope, showing spherical virions approximately 25–30 nm in diameter (Figure 2B).

### 3.3. Antigen Production of the Rescued O/Boeun/SKR/2017 Virus

The antigen yield was measured according to viral concentration and virus infection time. At 0.001 and 0.005 MOI, the maximum antigen levels were detected at 16 hpi, whereas at 0.01 and 0.05 MOI, the highest antigen levels were observed at 12 hpi (Figure 3A). The rescued O/Boeun/SKR/2017 virus derived from the infectious clone showed the highest antigen level (6.4 μg/mL) at 0.001 MOI after 16 hpi. Based on these results, we compared the time-dependent changes in viral titers and antigen levels between the rescued O/Boeun/SKR/2017 and parental O/Boeun/SKR/2017 at an MOI of 0.001. Both viruses showed the highest antigen levels (over 6 μg/mL) at 16 hpi, followed by a decrease thereafter (Figure 3B). The viral titers also peaked at 16 hpi and remained at similar levels thereafter (Figure 3C). The two viruses showed no statistically significant differences at any infection time.

### 3.4. Evaluation of the Protective Efficacy of the Rescued O/Boeun/SKR/2017 Virus

To evaluate whether the rescued O/Boeun/SKR/2017 virus could serve as a vaccine strain, it was inactivated and purified using sucrose density gradient ultracentrifugation. A test vaccine was prepared containing 15 μg per dose, and pigs were intramuscularly immunized. Four weeks post-vaccination, the pigs were challenged at the footpad with the O/Boeun/SKR/2017 virus. Sera were collected weekly to measure the antibody responses (Figure 4A). ELISA for structural protein antibodies showed that three pigs were seropositive at 28 d post-vaccination, immediately before the challenge (Figure 4B). Viral neutralization assays revealed that all animals developed neutralizing antibodies at levels just below 1/45 (Figure 4C). Upon viral challenge, all three pigs in the non-vaccinated control group exhibited clear clinical signs of FMD, and viral RNA was detected in the nasal swabs (Figure 4D). In contrast, none of the five vaccinated pigs displayed clinical symptoms, and no viral RNA was detected, indicating complete protection against FMDV challenge (Figure 4E).

## 4. Discussion

The construction of infectious FMDV clones has been previously reported by several researchers [20,21,22,23,24,25,26]. In this study, we constructed an infectious clone of O/Boeun/SKR/2017 based on the cloning method previously used for the full-length genome of the O1 Manisa strain [27]. While two fragments have been amplified by PCR using restriction enzymes, other researchers have divided the genome into four to five fragments when constructing infectious clones using restriction enzymes [21,22,24]. A unique feature of this cloning was the artificial creation of an AvrII recognition site by substituting the 375th base of the O/Boeun/SKR/2017 genome, which was used as the template, from adenine (A) to guanine (G) in the small-fragment reverse primer region for efficient cloning using restriction enzymes. While the restriction enzyme-based cloning approach requires searching for appropriate enzyme sites and occasionally altering some nucleotides, recent methods that use homologous recombination to clone more efficiently without restriction enzymes have been developed. Using this method, one report described the connection of four PCR-amplified fragments to construct an FMDV clone [23].

During the cloning process in this study, the amino acid at position 193 of VP2 changed from serine to glycine. However, the antigen yield of the rescued O/Boeun/SKR/2017 virus was almost the same as that of the parental O/Boeun/SKR/2017 virus. The viral titer also demonstrated a similar level, suggesting that this single amino acid substitution did not affect the replication of the rescued O/Boeun/SKR/2017 virus. However, previous reports indicate that when the 193rd residue of VP2 is substituted with tyrosine or phenylalanine, the heat stability of FMDV increases [28]. This suggests that depending on the amino acid present at the VP2 position, the physical properties of FMDV may vary.

While previous studies on infectious FMDV clones have mainly focused on viral pathogenicity, our study emphasizes the use of the infectious clone as a vaccine seed virus to evaluate antigen productivity and protective efficacy in animals after vaccination. The amount of FMD vaccine antigen is not measured by the concentration of individual structural proteins but rather by the concentration of intact 146S virus particles [29]. Once the virus particles dissociate into pentamers or smaller subunits, vaccine efficacy drops sharply [30]. Therefore, maximizing the 146S content is crucial for cost-effective vaccine production. Although exact antigen yields are usually proprietary to vaccine manufacturers, published data indicate that FMD vaccine antigens are typically recovered at concentrations of 0.3–3 μg/mL from viral supernatants: 0.3–0.4 µg/mL for serotypes O and A from an Indian vaccine plant [31]; 2.6 µg/mL for serotype O reported by Dutch researchers [28]; 1.6 µg/mL for recombinant SAT2 virus reported by South African and U.S. researchers [32]; 1.5 µg/mL for serotype O Campos virus, 0.6 µg/mL for serotype A, and 1.3 µg/mL for serotype C reported by Brazilian researchers [33]; and 2.5–3.1 µg/mL for serotype O reported by Chinese researchers [34]. Compared with these values, the 6.4 µg/mL antigen yield obtained in this study is considered exceptionally high. It has been reported that antigen content can vary even when viral titers are similar [35,36]. In our study, the antigen yield from the infectious clone–derived virus reached 6.4 μg/mL, significantly higher than in prior reports. This high productivity may reflect unique characteristics of the O/Boeun/SKR/2017 virus. Future studies comparing the amino acid sequences between highly productive and low-productive FMDV strains may identify the genetic regions responsible for antigen productivity. Previous studies cataloging viral mutations during the cell adaptation of FMDV have shown that the regions affecting viral adaptation to cell lines differ among viruses, making it challenging to establish a general rule [37].

It is not easy to predict virus protection solely based on neutralizing antibody titers without performing actual challenge experiments. Cut-off titers for evaluating immunological protection afforded by vaccination have to be established from the experience of potency test results with the relevant vaccine and target species [19]. However, according to a report [38], a neutralizing antibody titer in the range of 1.26–1.64 log_10_ is associated with approximately 75% protection. Based on this, the neutralizing titers observed in this study (1.2–1.5 log_10_) suggest a likely protective effect. Another study also reported virus protection in animals with VN titers as low as ≤1/45 [39].

The O/Boeun/SKR/2017 strain used in this study has been previously reported as a vaccine strain and has been successfully scaled up for production [36,40]. Due to its high antigen productivity, the O/Boeun/SKR/2017 virus is considered a valuable genetic backbone for the development of other FMD vaccine strains. Traditionally, the development of vaccine strains requires lengthy adaptation of field viruses to cell culture, with some viruses adapting poorly. Thus, replacing the P1 region of non-adapted field isolates with that of cell-adapted backbones has been proposed as a solution [20]. The infectious clone derived from the O/Boeun/SKR/2017 genome constructed in this study generated new infectious viruses within a few days of transfection into cells, making it a valuable platform for the rapid generation of diverse FMDV strains.

Furthermore, the construction of infectious clones allows the insertion of various molecular tags or markers. For example, FMDV structural proteins can be fused with His_6_ or HA tags for easy purification using affinity chromatography [41,42,43,44,45]. Infectious clones also enable insertion or deletion of specific markers to differentiate between vaccinated and field-infected animals. The FMDV nonstructural protein 3 B comprises three homologous repeats (3B1, 3B2, and 3B3), and antibody responses mainly target 3B1 or 3B2 [46,47]. Therefore, constructing recombinant viruses containing only 3B3 (with 3B1 and 3B2 deletions) results in vaccinated animals that test negative in the NSP antibody assays [48,49]. Numerous studies have reported improvements in viral stability through genetic modifications of infectious clones. The FMDV SAT and O types are known to be less stable than the others [50], and various attempts have been made to enhance their stability through gene replacement [3,28,51].

## 5. Conclusions

This study is the first to construct an infectious clone from a South Korean field isolate and to utilize it as a vaccine seed strain. As the rescued O/Boeun/SKR/2017 virus demonstrated exceptionally high antigen productivity, it may be possible that new vaccine strains with high antigen productivity could be developed more rapidly by replacing virus-neutralizing epitopes in infectious clones derived from the O/Boeun/SKR/2017 genome, potentially shortening the overall vaccine development timeline.

## Figures and Tables

**Figure 1 vaccines-13-01195-f001:**
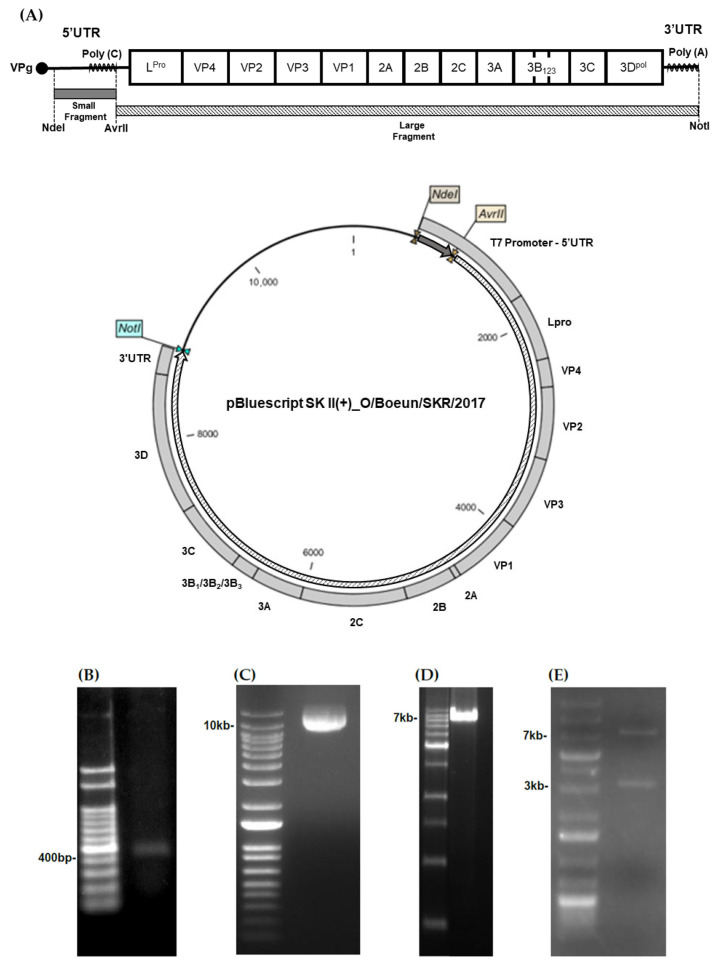
Construction of the O/Boeun/SKR/2017 infectious clone. (**A**) Schematic representation of the overall cloning strategy used to construct the full-length O/Boeun/SKR/2017 infectious clone. (**B**) The small genomic fragment (415 bp) of the O/Boeun/SKR/2017 strain was amplified by PCR and confirmed by agarose gel electrophoresis. (**C**) The plasmid harboring the full-length O1 Manisa genome was digested with AvrII (~11 kb). (**D**) The large genomic fragment (~7.8 kb) of the O/Boeun/SKR/2017 strain was amplified by PCR. (**E**) The large fragment of the O1 Manisa genome in the plasmid was substituted with that of the O/Boeun/SKR/2017 virus, and double digestion with AvrII and NotI confirmed the replacement.

**Figure 2 vaccines-13-01195-f002:**
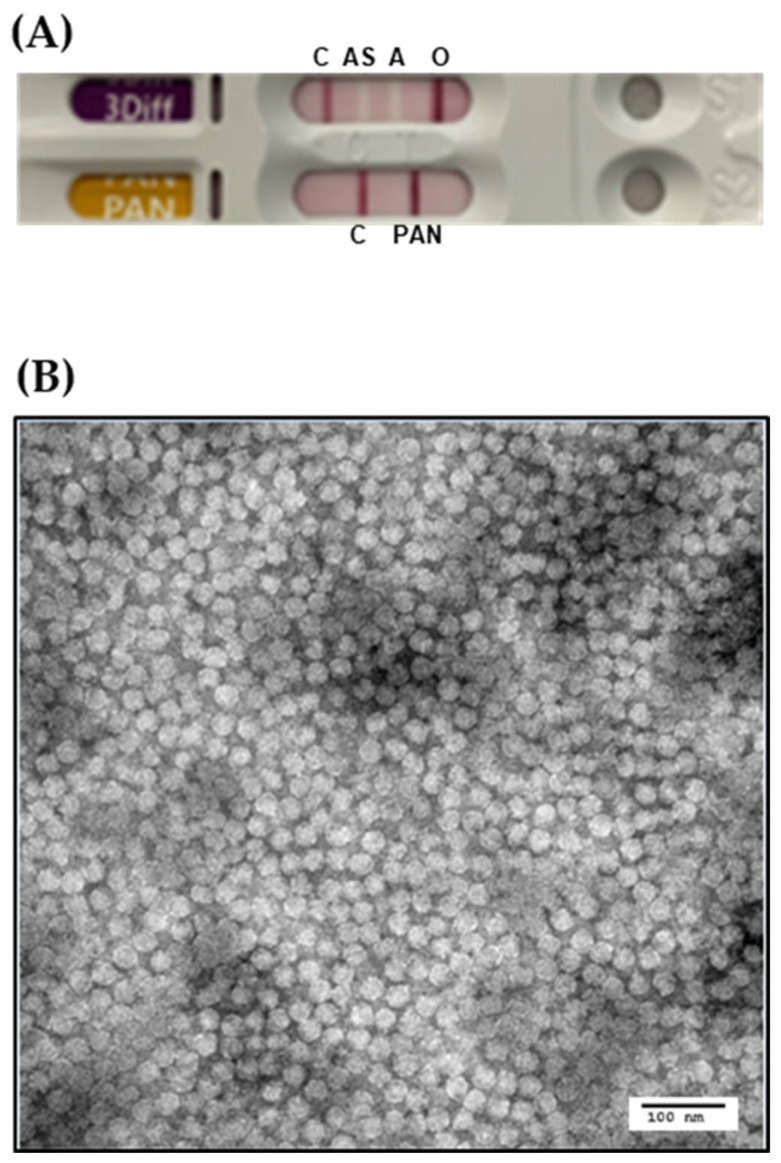
Serotyping and morphological characterization of the rescued O/Boeun/SKR/2017 virus. (**A**) Serotyping of the rescued FMDV was performed using the virus-infected culture supernatant and a commercial FMD antigen detection kit. (**B**) Transmission electron microscopy of virus particles purified by sucrose-gradient ultracentrifugation.

**Figure 3 vaccines-13-01195-f003:**
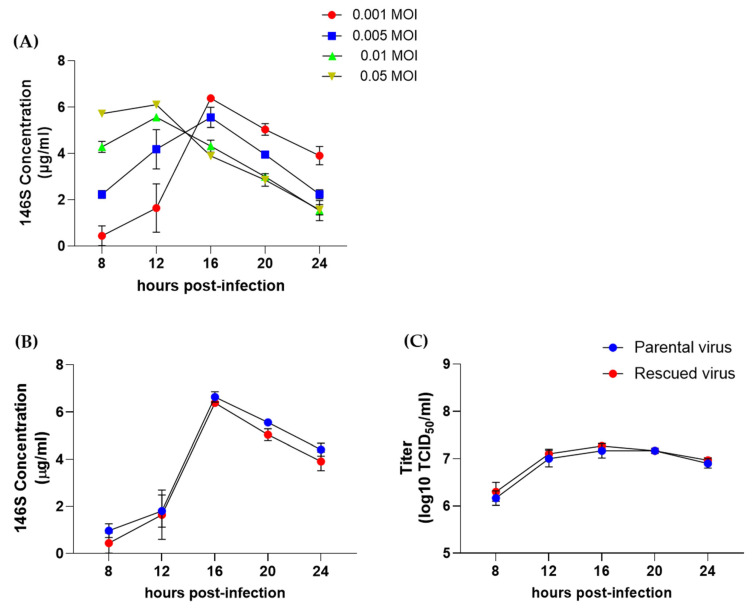
Growth kinetics of the rescued and parental O/Boeun/SKR/2017 viruses. (**A**) Antigen yield of the rescued virus under varying multiplicities of infection (MOI) and infection times. (**B**) Comparative antigen yields between the rescued and parental viruses at 0.001 MOI over time post-infection. (**C**) Comparison of viral titers between the rescued and parental viruses across different infection time points.

**Figure 4 vaccines-13-01195-f004:**
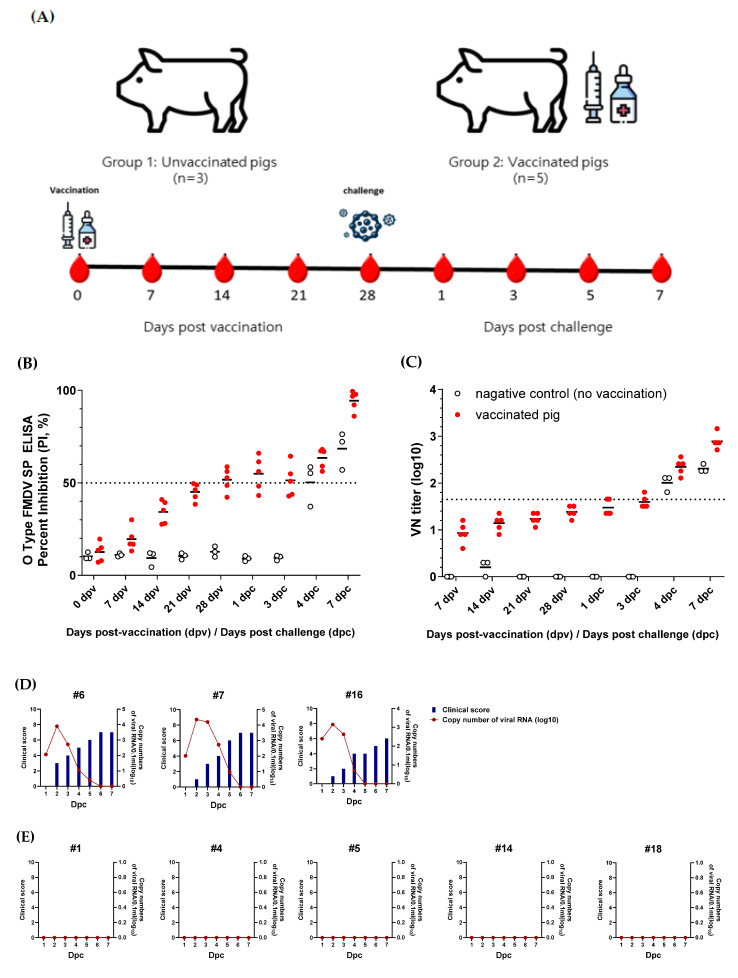
Protective efficacy of the rescued O/Boeun/SKR/2017 virus in pigs. Pigs were immunized with an inactivated vaccine derived from the rescued O/Boeun/SKR/2017 virus and subsequently challenged with the homologous field strain to evaluate protective immunity. (**A**) Schematic overview of the vaccination and challenge schedule, including serum collection intervals. (**B**) Structural-protein ELISA for sera collected weekly following vaccination and virus challenge. The dotted line indicates the threshold for a positive result. (**C**) virus-neutralization titers for sera collected weekly following vaccination and virus challenge. The dotted line indicates a titer of ≥1.65 log_10_. (**D**) Clinical scores and FMDV RNA detection in nasal swabs from unvaccinated control pigs. (**E**) Clinical scores and FMDV RNA detection in nasal swabs from vaccinated pigs post-virus challenge.

**Table 1 vaccines-13-01195-t001:** Primers employed for the amplification of the small and large genomic fragments used in constructing the O/Boeun/SKR/2017 infectious clone.

Region	Primer	Sequence (5′→3′)
Small fragment	Forward	GAC*CATATG*TAATACGACTCACTATAGGGTTGAAAGGGGGCGTTAGGGT
	Reverse	TTG*CCTAGG*GGGGGGGGGGGGGGGGGGTGAAAGGTGGGCTTC
Large fragment	Forward	TTG*CCTAGG*TTTTCCGTCGTCCCCGACGT
	Reverse	GAC*GCGGCCGC*TCTAGAACTAGTTTTTTTTTTTTTTTTTTTGGAAGAGGAAGCGGGAA

The T7 promoter sequence is underlined, and the restriction enzyme recognition sites are shown in italics.

## Data Availability

The original contributions presented in this study are included in the article. Further inquiries can be directed to the corresponding author.

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
