# Peer review of "Construction of a Full-Length Infectious Clone Derived from Type O Foot-and-Mouth Disease Virus Isolated in South Korea for Vaccine Development with High Antigen Productivity"

_vaccines, 2025, doi:10.3390/vaccines13121195_

Round 1
Reviewer 1 Report
Comments and Suggestions for Authors
This study successfully constructed a novel full-length infectious clone of the foot-and-mouth disease virus (O/Boeun/SKR/2017 strain) through genetic engineering and validated its potential for vaccine development, offering new strategies and approaches for future vaccine research.
Suggestions for revision:
1.The single amino acid change (Ser193 to Gly in VP2) is noted but not discussed in depth. Its potential impact on antigenicity or viral fitness should be addressed, even if no effect was observed.
2.The high antigen yield (6.4 µg/mL) is a key finding; more comparison with published yields and discussion of scalability would be valuable.
3.Adding commercially available vaccines as control groups in animal experiments allows for a more robust comparative evaluation of immunogenicity.
Author Response
Reviewer #1
- The single amino acid change (Ser193 to Gly in VP2) is noted but not discussed in depth. Its potential impact on antigenicity or viral fitness should be addressed, even if no effect was observed.
≫ In this study, although the 193rd amino acid residue of VP2 was unintentionally changed from serine to glycine during the gene-cloning step, this mutation did not affect the viral titer or antigen productivity of the FMDV. However, previous reports indicate that when the 193rd residue of VP2 is substituted with tyrosine or phenylalanine, the heat stability of FMDV increases [28]. This suggests that depending on the amino acid present at the VP2 position, the physical properties of FMDV may vary.
This information has been added to the Discussion section (line 361-364).
- The high antigen yield (6.4 µg/mL) is a key finding; more comparison with published yields and discussion of scalability would be valuable.
≫ Antigen yields reported by other research groups include: 0.3–0.4 µg/mL for serotypes O and A from an Indian vaccine plant [31]; 2.6 µg/mL for serotype O reported by Dutch researchers [28]; 1.6 µg/mL for recombinant SAT2 virus reported by South African and U.S. researchers [32]; 1.5 µg/mL for serotype O Campos virus, 0.6 µg/mL for serotype A, and 1.3 µg/mL for serotype C reported by Brazilian researchers [33]; and 2.5–3.1 µg/mL for serotype O reported by Chinese researchers [34]. Compared with these values, the 6.4 µg/mL antigen yield obtained in this study is considered exceptionally high.
This information was added to the Discussion section (line 374-380).
- Adding commercially available vaccines as control groups in animal experiments allows for a more robust comparative evaluation of immunogenicity.
≫ To examine whether viral protection by vaccination was induced in pigs, the results were compared with a non-vaccinated control group, which served as a positive control for clinical disease development. Protection against viral challenge can be accurately determined by comparison with the non-vaccinated control group. Therefore, comparison with a licensed commercial vaccine—already proven to confer protection—was unnecessary.
Reviewer 2 Report
Comments and Suggestions for Authors
This manuscript presents the construction and evaluation of an infectious clone derived from S.K FMDV O, including its use as a vaccine strain. This study provides a valuable platform for rapid vaccine development. There are few missing or underdeveloped details in the MM, result and conclusion sections that needs to be addresses please.
- It lacks a plasmid map or schematic which is essential for understanding the cloning strategy and restriction sites.
- The purpose and method of the reported Ser to Gly substitution at the VP2 position 193 is unclear.
- There is no mention of full genome sequencing to confirm the integrity of the recombinant plasmid.
- Details of plasmid concentration, purity and verification methods prior to transfection are missing.
- the transfection SOP has missing parameters like DNA quantity, cell density and transfection efficacy.
- No quantitative data is provided to support virus rescue and propagation claims like TCID50 or plaque assay.
- There is no negative controls or replicates are described.
- Quantification of 146S particles lacks validation details such as standard curve, sensitivity and specificity.
- The manuscript does not include ethical approval statement and no description of randomization or statistical analysis for the animal study.
- The details of group sizes ( 5 vaccinated, 3 controls) is not provided.
- ELISA and VNT are described but cutoff values, interpretation criteria and replicate numbers are missing.
- VN titers are reported at just below 1/45 without context, please clarify whether this meets protective thresholds.
- The ANOVA and Tukey's HSD test does not specify software version.
- The conclusion claim of exceptionally high antigen productivity should be qualified with comparative data or benchmarks.
- The statement about rapid development and shortening the vaccine timeline are speculative and should be framed as potential applications, not proven outcomes.
It is recommended that addressing these points will significantly enhance the clarity, reproducibility and impact of this study please. Thank you.
Author Response
Reviewer #2
- It lacks a plasmid map or schematic which is essential for understanding the cloning strategy and restriction sites.
≫ A plasmid map was drawn and added to Figure 1(a) as requested.
- The purpose and method of the reported Ser to Gly substitution at the VP2 position 193 is unclear.
≫ The amino acid substitution at VP2 position 193 (serine to glycine) was not intentional but occurred during the gene amplification process in this study.
- There is no mention of full genome sequencing to confirm the integrity of the recombinant plasmid.
≫ As pointed out by the reviewer, we indicated in the manuscript that full-length sequencing analysis was performed (line 267-268).
- Details of plasmid concentration, purity and verification methods prior to transfection are missing.
≫ The following information has been added to the manuscript (line 151–156).
The plasmid was purified using the Nucleo Spin Plasmid Kit (MACHEREY-NAGEL, Düren, Germany). Following purification, plasmid concentration and purity were assessed using a DeNovix DS-11 Fx+ spectrophotometer (DeNovix, Inc., Delaware, USA). The results showed an A260/280 ratio of 1.85 and an A260/230 ratio of 2.18, confirming that the plasmid DNA was of high purity with no detectable protein or organic compound contamination.
- the transfection SOP has missing parameters like DNA quantity, cell density and transfection efficacy.
≫ The following information has been added to the manuscript (line 156–159).
The purified recombinant plasmid was then transfected into BHK/T7-9 cells expressing T7 RNA polymerase using Lipofectamine 3000 (Invitrogen, Waltham, MA, USA), according to the manufacturer’s protocol. For transfection, 1 × 10⁶ BHK/T7-9 cells were seeded in a 6-well plate, and 2 µg of the O/Boeun/SKR/2017 expression vector was applied.
- No quantitative data is provided to support virus rescue and propagation claims like TCID50 or plaque assay.
≫ As suggested by the reviewer, the viral titer (TCID₅₀) after CPE has been included in the main text (line 287–289).
The viral titer of the harvested supernatant was determined by TCID₅₀ assay, yielding a titer of 1.28 × 10⁸ TCID₅₀/mL
- There is no negative controls or replicates are described.
≫ Because infectious virus was detected after transfecting cells with the infectious clone and FMDV recovery was confirmed by an antigen detection kit, electron microscopy, and sequencing analysis there was no need for a negative control group.
- Quantification of 146S particles lacks validation details such as standard curve, sensitivity and specificity.
≫ The following information has been added to the Materials and Methods section (line 183-184).
The FMDV particle-quantification method has already been described in previous studies [15,16]. therefore, the detailed procedure was omitted in this manuscript.
- The manuscript does not include ethical approval statement and no description of randomization or statistical analysis for the animal study.
≫ The following information has been added to the Materials and Methods section under Animal Experiments (line 224-226).
Animal experiments were approved by the Institutional Animal Care and Use Committee (IACUC) of the Animal and Plant Quarantine Agency (IACUC No. 2024-831) on February 8, 2024.
- The details of group sizes (5 vaccinated, 3 controls) is not provided.
≫ As requested, detailed information on experimental animal groups has been added (line 208-214).
A minimum of three animals is required in both the experimental and control groups to allow for basic statistical analysis; therefore, the negative control group was set at three animals. In vaccine studies, unexpected mortality or dropout may occur due to stress during vaccination or virus challenge, as well as individual variability. Because the loss of even a single animal in the experimental group can substantially affect the statistical interpretation, the vaccination group was composed of five animals to provide an adequate margin.
- ELISA and VNT are described but cutoff values, interpretation criteria and replicate numbers are missing.
≫ The criteria for positive/negative determination for both assays and the number of experimental replicates have been added to the manuscript (line 247-249 & line 237-240).
The ELISA results were interpreted based on percentage inhibition, with values ≥ 50% considered positive and those < 50% considered negative
In general, a VN titer of 1/45 or more of the final serum dilution is regarded as positive. However, for certification of individual animal for the purpose of international trade, a titer of 1/16 is considered to be positive (WOAH).
- VN titers are reported at just below 1/45 without context, please clarify whether this meets protective thresholds.
≫ It is not easy to predict virus protection solely based on neutralizing antibody titers without performing actual challenge experiments. Cut-off titers for evaluating immunological protection afforded by vaccination have to be established from experience of potency test results with the relevant vaccine and target species [19]. However, according to a report [38], a neutralizing antibody titer in the range of 1.26–1.64 log₁₀ is associated with approximately 75% protection. Based on this, the neutralizing titers observed in this study (1.2–1.5 log₁₀) suggest a likely protective effect. Another study also reported virus protection in animals with VN titers as low as ≤1/45 [39].
This information has been added to the manuscript (line 390-397).
- The ANOVA and Tukey's HSD test does not specify software version.
Those are the built-in function in Prism 8 software
- The conclusion claim of exceptionally high antigen productivity should be qualified with comparative data or benchmarks.
≫ Antigen yields reported by other research groups include: 0.3–0.4 µg/mL for serotypes O and A from an Indian vaccine plant [31]; 2.6 µg/mL for serotype O reported by Dutch researchers [28]; 1.6 µg/mL for recombinant SAT2 virus reported by South African and U.S. researchers [32]; 1.5 µg/mL for serotype O Campos virus, 0.6 µg/mL for serotype A, and 1.3 µg/mL for serotype C reported by Brazilian researchers [33]; and 2.5–3.1 µg/mL for serotype O reported by Chinese researchers [34]. Compared with these values, the 6.4 µg/mL antigen yield obtained in this study is considered exceptionally high.
This information was added to the Discussion section (line 374-380).
- The statement about rapid development and shortening the vaccine timeline are speculative and should be framed as potential applications, not proven outcomes.
≫ The sentence has been revised according to the reviewer’s comment (line 423-427).
It may be possible that new vaccine strains with high antigen productivity could be developed more rapidly by replacing virus neutralizing epitopes in infectious clones derived from the O/Boeun/SKR/2017 genome, potentially shortening the overall vaccine development timeline.
Round 2
Reviewer 1 Report
Comments and Suggestions for Authors
I have no further comments.